# Strange Hadrons in Underlying-Events as Measured in Proton-Proton Collisions at $\sqrt{s} = 13$ TeV by the ATLAS Detector at LHC

**F. Djama on behalf of the ATLAS Collaboration**

Aix-Marseille Université, CNRS/IN2P3, CPPM Marseille
djama@cppm.in2p3.fr ,

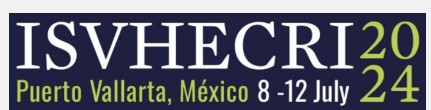

*22nd International Symposium on Very High Energy Cosmic Ray Interactions (ISVHECRI 2024) Puerto Vallarta, Mexico, 8-12 July 2024* doi:10.21468/SciPostPhysProc.?

## Abstract

Recent measurements of distributions sensitive to the underlying-event, the hadronic activity observed in relationship with the hard scattering in the event, by the ATLAS experiment are presented. Used data were recorded during the LHC collider Run II at $\sqrt{s}$ = 13 TeV. Average particle multiplicity are measured for $K_S^0$ and $\Lambda$ baryons as a function of the leading track-jet transverse momentum and total underlying event multiplicity, and are compared to Monte Carlo simulation predictions which in general fail to describe the data.

# 1 Introduction

Underlying-Event (UE) in hadronic collisions arises from initial and final state radiation, color reconnection, Multi-Parton Interactions (MPI) and beam remnants in case of diffractive scattering. UE is important for a correct modeling of proton-proton collisions and their extrapolation to hadron-hadron collisions, especially for studies of air-shower triggered by high energy cosmic rays. Perturbative QCD (pQCD) computes the partonic hard scattering but is unable to describe hadronisation and UE. Precise measurements on UE are of peculiar importance to feed models of hadronic collisions and improve our understanding of strong interactions in their non-perturbative regime.

Strange particles production is a probe to investigate UE. The mass of strange quark $m_s$ is of the same order of magnitude than the QCD scale factor $\Lambda_{QCD}$, which makes pQCD computation ineffective in strange quark production. $K_S^0$ and $\Lambda$ are the most accessible strange particles since they are the lightest strange meson and baryon respectively. They are also easy to tag, by their displaced decay vertex $V^0$ into two charged particles $K_S^0 \rightarrow \pi^+\pi^-$, $\Lambda \rightarrow \pi^- p$ and $\bar{\Lambda} \rightarrow \pi^+\bar{p}$. The $c\tau$ of these decays are respectively 2.7 and 7.9 cm, which ensures the $V^0$ to be reconstructed in the volume of the inner tracker of the ATLAS experiment.

This contribution is about the production of $K_S^0$ and $\Lambda$ in UE of proton-proton collisions at $\sqrt{s} = 13$ TeV, as measured in the ATLAS detector [1] at the LHC collider [2]. The full analysis is described in [3]. A similar measurement has been achieved at $\sqrt{s} = 7$ TeV by the CMS Collaboration in 2013 [4].

# 2 Experimental Setup and Data

A study of UE should be carried out on a non-biased sample of proton-proton collisions. But even events acquired with minimum bias triggers are biased because they contain particles which have transverse momentum $p_T$ slightly larger than average particles of UE. The CDF Collaboration at the Tevatron proposed a method to get rid of this bias [5]. The leading-jet $p_T$ is identified and is considered as the direct product of the hard scatter. All particles within $\pi/3$ in azimuth from the leading-jet direction (Towards region) and within the same angular distance from the azimuth of the back-to-back to leading-jet direction (Away zone) are excluded from UE. Only particles belonging to the region between the Towards and Away regions are considered as part of UE. This region is called the Transverse region.

Pileup is likely to destroy this definition of UE and the events should be collected in special low luminosity fills of the LHC collider.

The ATLAS Experiment [1] has a precise silicon pixel detector to reconstruct primary and secondary vertices. The pixel detector has four barrel layers located at radii of about 3.5, 5.5, 9 and 12.5 cm and is closed by three pixel disks on both endcaps. It is completed at higher radii by four double layers of silicon microstrip sensors and gaseous straw tubes. Both microstrips and straw tubes have extensions in the endcaps in order to achieve a precise charged particle tracking volume in a pseudorapidity domain of $|\eta| < 2.5$. Pixel, microstrip and straw tubes detectors form the ATLAS tracker called Inner Detector (ID).

Twelve sectors of Minimum Bias Trigger Scintillators (MBTS) are located at distances from

the interaction point (IP) of $z = \pm 3.56$ m and cover pseudorapidities $|\eta|$ between 2.07 and 3.86. The twelve sectors are arranged in two radial rings separated at $|\eta| = 2.76$. The outer and inner rings have four and eight azimuthal sectors respectively. ATLAS has calormieters, muon systems and forward detectors which are not used in this analysis.

Six special LHC fills have been recorded by ATLAS during June 2015. Only 29 colliding bunches were present in order to get rid of pileup. Obtained mean number of inelastic collisions per bunch crossing was between 0.003 and 0.03, while it was routinely above 20 in regular LHC fills. Smaller bunches and larger betatronic function contributed also to achieve low luminosity. A very simple minimum bias trigger requiring at least a single MBTS sector above threshold has been used for five fills. At least one such sector on both sides of IP were required for the sixth fill. About 100 and 20 million minimum bias events were recorded with these two triggers, respectively.

## 3   Simulation Samples

Three Monte Carlo (MC) generators are used, EPOS 3.4 [6] and two PYTHIA8 [7] tunes.

EPOS 3.4 with EPOS-LHC tune [8] is based on Gribov-Regge theory [9] where elementary interactions occur by multi-pomeron exchange, while hadronisation takes into account hydrodynamical nuclear effects.

PYTHIA8 uses t-channel gluon and pomeron exchange for inelastic and diffraction processes respectively. MPIs are included by an impact parameter approach, while hadronisation is achieved by the Lund model [10]. Two PYTHIA8 tunes are used: A.2 tune, using MSTW2008 LO parton distribution functions (pdf) [11] and tuned to ATLAS minimum bias events at $\sqrt{s} = 7$ TeV, and Monash tune [12], using NNPDF2.3 LO pdf [13], tuned on SPS, LHC and Tevatron data. For this analysis, an alternative color reconnection model [14] has been added to the Monash tune.

The response of the ATLAS detector to EPOS-LHC and PYTHIA8-A2 events was simulated by GEANT4 [15] [16], while PYTHIA8-Monash events were used at particle level.

## 4   Prompt Tracks and Jet Selection

To identify the leading-jet in each event, prompt charged particles and hadronic jets have to be selected. Charged particle tracks are reconstructed by the ID. Prompt tracks are required to have a $p_{\mathrm{T}}$ larger than 500 MeV, a pseudorapidity $|\eta| < 2.5$, and transverse and longitudinal impact parameters smaller than 1.5 mm. To ensure good track quality, minimum hits on silicon detectors and maximal cut on the $\chi^2$ track fit are added.

Hadronic jets are identified using the anti-$k_t$ algorithm [17] with $\Delta R = 0.4$. To ensure full jet containment in the tracking volume, the jet pseudorapidity is required to be $|\eta| < 2.1$.

## 5   $K_S^0$ and $\Lambda$ Identification

To reconstruct $V^0$, large radius tracks are allowed in the pattern recognition and track fit procedures. Only non-used hits by the prompt tracks are considered. Large radius tracks are then selected like prompt ones, with loosened impact parameter criterias.

$V^0$ are found by iterating over all possible pairs of oppositely charged tracks using both prompt and large radius tracks. Each found vertex is then fitted with the three following hypothesis: $K_S^0 \to \pi^+\pi^-$, $\Lambda \to \pi^- p$ and $\bar{\Lambda} \to \pi^+ \bar{p}$. Selected vertices must have their invariant

mass in agreement with one hypothesis: $|M_{V^0} - M_{K_S^0}| < 20$ MeV or $|M_{V^0} - M_\Lambda| < 7$ MeV. Quality cuts on $\eta$, $p_T$, direction, decay length, mass uncertainty and minimal distance between same species $V^0$ are added. Each selected $V^0$ must satisfy one and only one hypothesis.

# 6   Event Selection

Data and MC events are required to have a primary vertex from at least two tracks with $p_T$ > 100 MeV, and at least one prompt track with $p_T$ > 1 GeV. Pile-up events are rejected. Numbers of events are corrected for trigger efficiency. Numbers of prompt tracks and $V^0$ are corrected for reconstruction efficiency, while the estimated fake $V^0$ are subtracted from the later.

An unfolding procedure using EPOS-LHC simulation has been used to account for migration between $p_T$ bins of reconstructed jets.

# 7   Systematic Uncertainties

Corrected numbers of events, tracks and vertices are subject to systematics uncertainties related to selection and correction procedures.

Unfolding uncertainty has been estimated by applying the procedure to EPOS-LHC simulated events. An alternative MC simulation (PYTHIA8-A2) has been used to derive uncertainty of unfolding model.

Difference in $V^0$ fake rate between data and simulation, material budget and statistical errors contribute to uncertainty on strange hadron rates. Similarly, material budget and track selection uncertainties are assigned to numbers of prompt tracks.

Systematic uncertainties associated with residual non-matching between reconstruction and particle level after unfolding, and $V^0$ efficiency correction estimated at particle level and applied on reconstructed data, have also been estimated.

The break down of systematics uncertainties on $K_S^0$ rate as function of the event leading-jet $p_T$ in the Transverse region is shown on Figure 1.

# 8   Results

In order to ensure the least possible biased data, the LHC fill recorded with the two hemisphere trigger is not used.

Figures 2a, 2c and 2e show the $K_S^0$ rate dependence on leading-jet $p_T$ in the three regions, as measured in data and predicted by the three flavour MC simulation. All regions show an increasing soft and a rather constant hard regimes, with a transition at a leading-jet $p_T$ of about 10 GeV. EPOS-LHC is the closest to data in soft regime, while it shows a dip in hard regime, in desagreement with data. PYTHIA8-Monash is better in the Towards region, while PHYTIA8-A2 models well the data shape but misses scale.

Figures 2b, 2d and 2f show the $K_S^0$ rate normalised to prompt particles. Soft to hard transition is less distinct. All models reproduce data shape. PYTHIA8-Monash is again better in the Towards region while EPOS-LHC does not perform as well as in event normalised rate.

Figure 3 shows the same distributions as Figure 2, but for $\Lambda + \bar\Lambda$ rates. Comments made above on Figures 2a, 2c and 2e, apply also on Figures 3a, 3c and 3e, except that this time, PYTHIA8-Monash reproduces data in all regions. For $\Lambda$ rate normalised to prompt particles, EPOS-LHC is still the closest to data in the soft regime and reproduces them well in the hard regime. PYTHIA8-Monash reproduces the shape in all regions and the scale in Away and

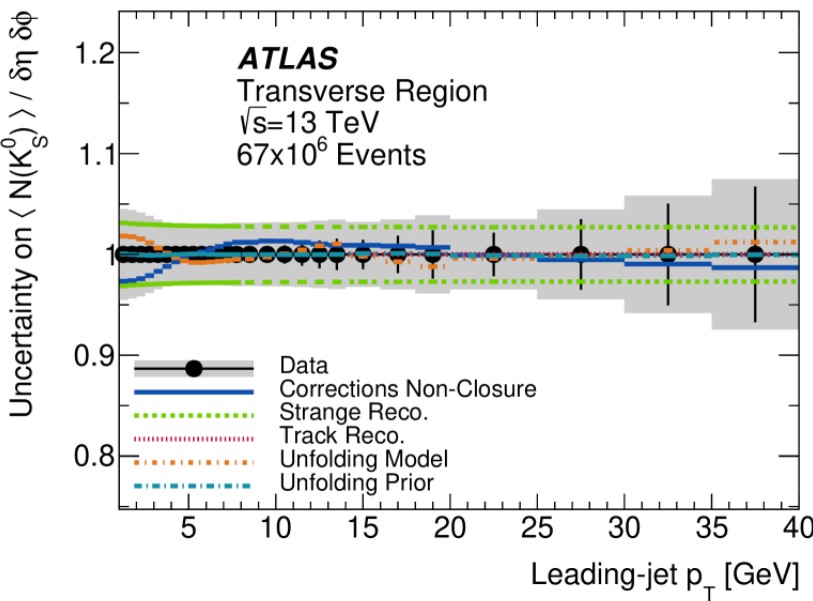

Figure 1: Breakdown of uncertainties in the measurement of the event-normalised mean number of $K_S^0$ in the Transverse region as a function of leading-jet $p_T$. Error bars show the statistical error and the shaded bands show the total uncertainty. The figure is extracted from [3].

Transverse regions. PYTHIA8-A2 misses the data scale by almost a factor two in all three regions.

The increasing of strange hadron yield in the soft regime confirms the impact parameter b picture of MPI: Higher $p_T$ leading-jet leads to smaller b and so to larger MPI. The yield then saturates for totally central collisions. Strange hadron yield normalised by prompt charged particles varies much less with leading-jet $p_T$ than when normalised by event: MPI is independent from hadronisation. The EPOS-LHC model is better at soft than at hard regime: It needs a better modeling for hard processes.

To study specifically the hard regime, the LHC fill recorded with the double hemisphere trigger was added to increase the statistics. The introduced bias on events where the leading-jet has $p_T$ higher than 10 GeV was found totally negligible.

Figures 4a and 4b show respectively the number of $K_S^0$ and $\Lambda + \bar{\Lambda}$ divided by the number of prompt particles in the Towards region as a function of the number of prompt particles in the Transverse region. The later quantity is used as a proxy to MPI activity. EPOS-LHC is the closest to data. PYTHIA8-Monash predicts a flat $K_S^0$ yields, in contradiction with data, while PYTHIA8-A2 is yet missing the data scale. The dependance of the ratio $N(\Lambda + \bar{\Lambda})/N(K_S^0)$ on number of prompt charged particles in the Transverse region is shown in Figures 4c and 4d for Towards and Transverse region respectively. PYTHIA8-A2 model correctly predicts the independant behaviour of this ratio in Transverse region but once again, does not reproduce data scale.

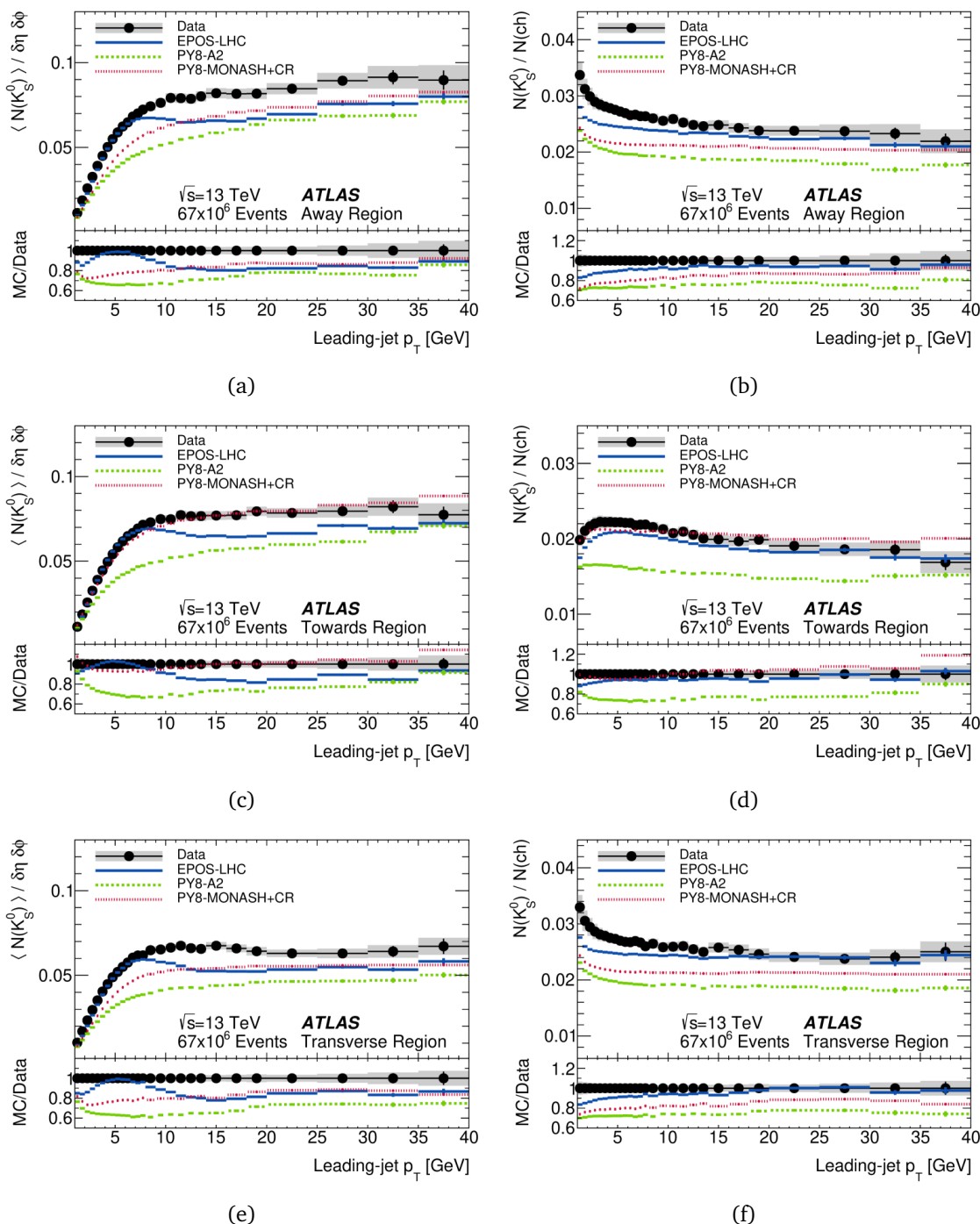

Figure 2: (Left) Per event and per unit $(\eta, \phi)$ normalised and (right) prompt charged-particle normalised $K_S^0$ yields as a function of leading-jet $p_T$ in the (a, b) Away, (c, d) Towards and (e, f) Transverse regions. Error bars show the statistical error and the shaded bands show the total uncertainty. The figure is extracted from [3].

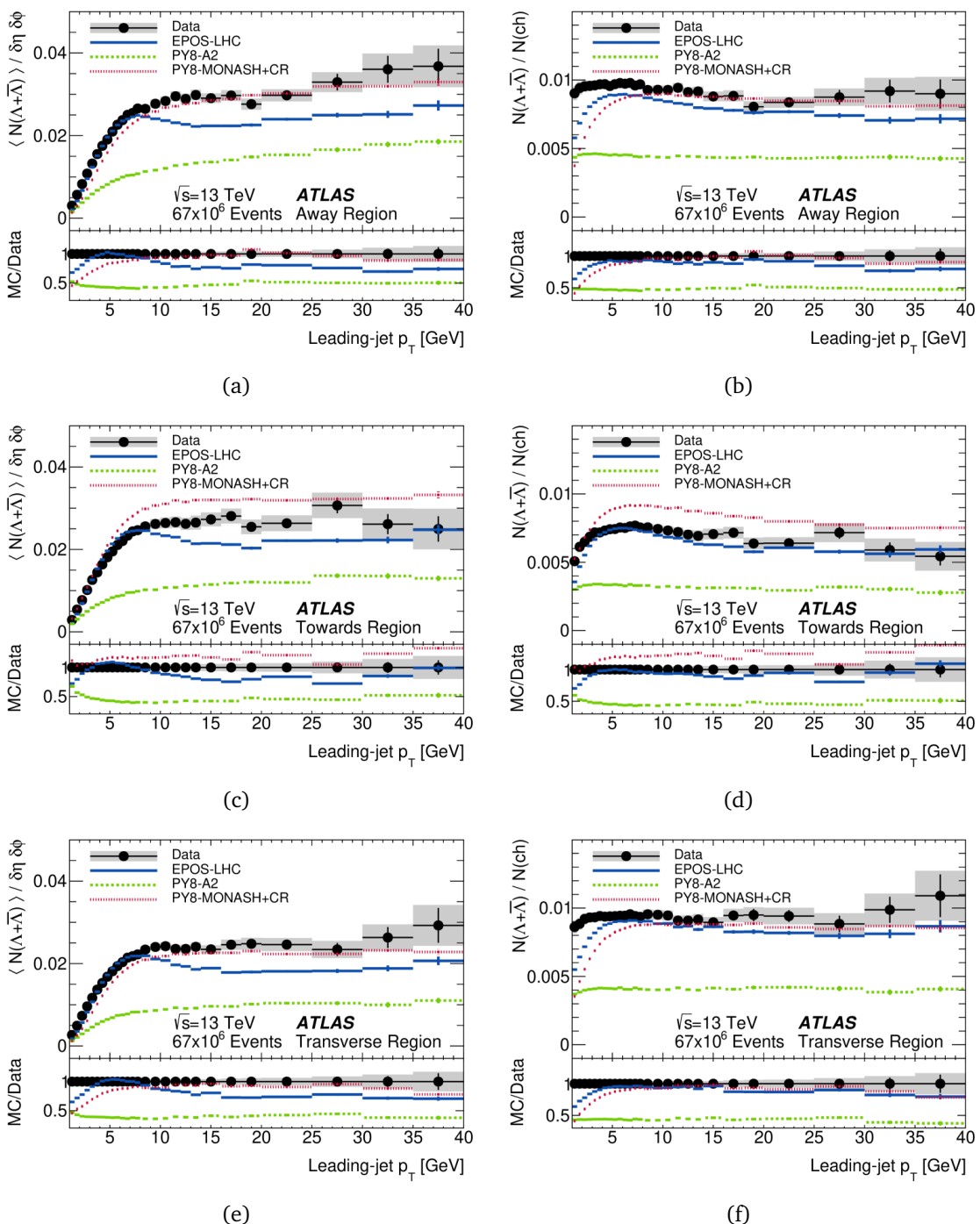

Figure 3: (Left) Per event and per unit $(\eta, \phi)$ normalised and (right) prompt charged-particle normalised $(\Lambda + \bar{\Lambda})$ yields as a function of leading-jet $p_T$ in the (a, b) Away, (c, d) Towards and (e, f) Transverse regions. Error bars show the statistical error and the shaded bands show the total uncertainty. The figure is extracted from [3].

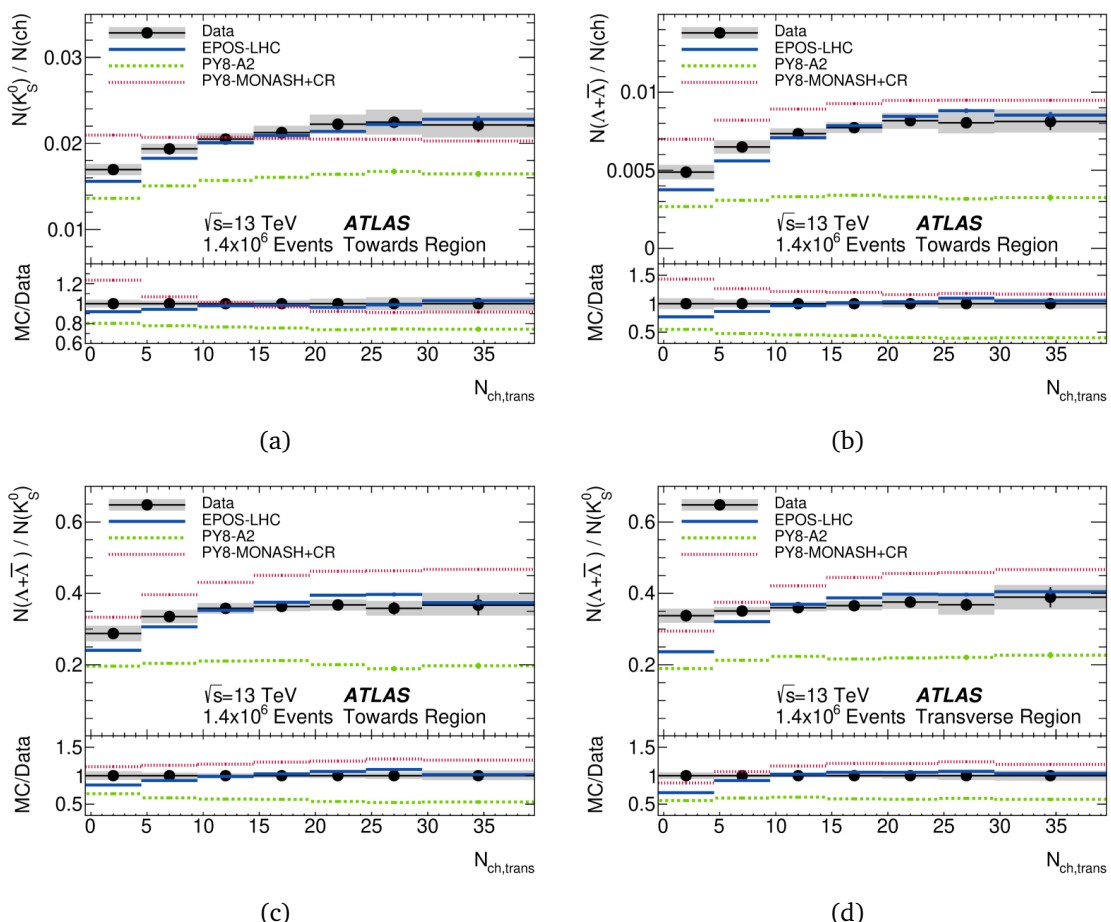

Figure 4: Comparison between data and MC simulation for several multiplicity ratios as a function of $N_{ch,trans}$ in events with leading jet $10 < p_T < 40$ GeV. Shown are the prompt charged-particle normalised (a) $K_S^0$ and (b) $(\Lambda + \bar{\Lambda})$ multiplicity yields in the Towards region, and relative yields of $(\Lambda + \bar{\Lambda})$ to $K_S^0$ in the (c) Towards and (d) Transverse regions. Error bars show the statistical error and the shaded bands show the total uncertainty. The figure is extracted from [3].

## 9   Conclusions

Properties of underlying-event were investigated with the strange hadrons $K_S^0$, $\Lambda$ and $\bar{\Lambda}$ in proton-proton collisions at $\sqrt{s} = 13$ TeV. Strange hadrons were identified via their decay secondary vertices. Their multiplicities and multiplicity ratios were compared to different Monte Carlo simulations. PYTHIA8-A2 underestimates the strange hadron yields by 40 to 50 %. PYTHIA8-Monash with the alternative colour reconnection scheme is much closer to data. EPOS-LHC is the best at soft regime and underestimates the yields at higher $p_T$. EPOS-LHC is in best agreement for yields variations with the number of prompt charged particle in the Transverse region.

These results may be used to improve the modeling of non-perturbative effects in simulations. More results are to come on hadronic interactions from ATLAS Collaboration.

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
