# Peer review of "Strange Hadrons in Underlying-Events as Measured in Proton-Proton Collisions at sqrt(s) = 13 TeV by the ATLAS Detector at LHC"

_SciPost Physics Proceedings_

## Round 1 · Referee Report · Anonymous (Referee 1) · 2025-10-22

Strengths

This manuscript describes productions of strangeness particles in under-lying events, including the forward region of ATLAS at 13 TeV pp collisions with a special low luminosity operation of LHC. Strange particles in the high-energy air showers are relevant to atmospheric lepton components. However, it is poorly known both experimentally and theoretically at the LHC energy, and thus, this measurement is unique and important.

Weaknesses

It may be improved to include more description for how much impact of this measurement on cosmic ray physics in the Introduction section, as this manuscript is for ISVHECRI proceedings, where high-energy cosmic ray interactions matter.

Report

This manuscript describes productions of strangeness particles in under-lying events including the forward region of ATLAS at 13 TeV pp collisions with a special low luminosity operation of LHC. Strange particles in the high energy air showers are relevant to atmospheric lepton components. However, it is poorly known both experimentally and theoretically at the LHC energy, and thus this measurement is unique and important. Manuscript is well-written with plots taken from already accepted for journal publication. Throughout the manuscript, no typo seems to be found. Therefore, I recommend approving this without any modification. It may be improved readability for the readers to include more description in the introduction section for how much impact on the cosmic ray physics as this manuscript is for ISVHECRI proceedings where high-energy cosmic ray interaction is the subject.

Requested changes

No change is mandatory for acceptance.

Recommendation

Publish (surpasses expectations and criteria for this Journal; among top 10%)

---

## Editorial Decision

accepted_in_target_journal